# Brain–Periphery Axes: The Potential Role of Extracellular Vesicles-Delivered miRNAs

**DOI:** 10.3390/biology13121056

**Published:** 2024-12-16

**Authors:** Giuseppa D’Amico, Adelaide Carista, Olga Maria Manna, Letizia Paladino, Domiziana Picone, Silvia Sarullo, Martina Sausa, Francesco Cappello, Alessandra Maria Vitale, Celeste Caruso Bavisotto

**Affiliations:** 1Department of Biomedicine, Neurosciences and Advanced Diagnostics (BiND), University of Palermo, 90127 Palermo, Italy; giuseppa.damico01@unipa.it (G.D.); adelaide.carista@unipa.it (A.C.); olgamaria.manna@unipa.it (O.M.M.); letizia.paladino@unipa.it (L.P.); domiziana.picone@unipa.it (D.P.); silvia.sarullo@unipa.it (S.S.); francesco.cappello@unipa.it (F.C.); celeste.carusobavisotto@unipa.it (C.C.B.); 2Euro-Mediterranean Institute of Science and Technology (IEMEST), Via Michele Miraglia 20, 90139 Palermo, Italy; 3Department of Theoretical and Applied Sciences, eCampus University, 22060 Novedrate, Italy; martina.sausa@studenti.uniecampus.it

**Keywords:** brain–periphery bidirectional communication, extracellular vesicles-delivered miRNAs, brain, gut, lung, heart, skeletal muscle, adipose tissue, kidney, immune system

## Abstract

Numerous findings demonstrate that the central nervous system and the peripheral organs and tissues communicate in physiological and pathological conditions thanks to the exchange of signaling molecules transported via blood, cerebrospinal fluid, and lymph. Extracellular vesicles, which can cross tissue barriers, including the blood–brain barrier, play a key role in this bidirectional communication by safely delivering these molecules, including proteins, lipids, and nucleic acids such as microRNAs (miRNAs). In particular, the latter involved in the post-transcriptional regulation of gene expression can alter recipient cells’ functions and orchestrate their behavior in response to different biological events. Thus, elucidating the role played by EVs-delivered miRNAs in brain–periphery communication could offer valuable insights for advancing the treatment and management of neurological and systemic diseases.

## 1. Introduction

MicroRNAs (miRNAs), the most abundant class of small noncoding RNAs (ncRNAs), are short (18–24 nucleotides), single stranded RNAs that originate from the highly regulated processing of longer polycistronic primary transcripts (pri-miRNAs). The pri-miRNA is firstly cut, within the nucleus co- or post-transcriptionally, to generate a hairpin-shaped precursor RNA (pre-miRNA) that is exported to the cytoplasm, where a second cleavage event takes place, producing a shorter double-stranded miRNA. Ultimately, one of the strands forms the mature miRNA, which is incorporated into the miRNA-induced silencing complex (miRISC). This complex specifically recognizes and binds to the 3′ UTR of a target mRNA, interfering with its stability or translation [1]. Therefore, miRNAs usually perform a negative post-transcriptional regulation of gene expression [2]. However, over the years, different complementary target sequences have been identified in other regions of the transcript, including the 5′ UTR and the coding sequence, as well as within promoter regions, with the description of several other mechanisms of action and effects on gene expression. Usually, the binding to 5′ UTR and to the coding region have silencing effects on gene expression while miRNA interaction with promoter region has been reported to induce transcription (Table 1) [3,4,5,6,7].

Thanks to these regulatory activities, miRNAs play a critical role in normal animal development and are involved in a variety of biological processes [8]. In humans, over 2000 miRNAs have been annotated and validated, which regulate an even greater number of proteins encoding genes, and, presumably, all biological events [9]. Therefore, variation of their expression patterns may result in different pathological conditions [10,11,12,13,14]. These findings confer them a reliable diagnostic and predictive/prognostic value, for assessing and predicting clinical outcomes. Furthermore, they could be valuable therapeutic agents/targets for developing personalized treatments, although miRNA-based therapies have not yet been incorporated into current clinical practice [15,16,17].

In this regard, behind tissue-specific miRNAs that form different clusters specifically expressed across different human tissues [18], great interest has elicited the less abundant and highly stable circulating miRNAs, i.e., those miRNAs found in biofluid such as serum, plasma, saliva, urine, either as extracellular vesicles (EVs)-encapsulated molecules, or as EVs-free and AGO proteins-associated molecules [16,17,19,20]. However, AGO proteins-associated miRNAs are released non-selectively after parental cell injury or death and are not up-taken by other cells. In contrast, the release of EVs-associated miRNAs is a regulated process, and these miRNAs can be transferred to recipient cells and alter their gene expression, mediating intercellular and inter-organ communication both in physiological and pathological conditions [21,22,23,24,25,26,27,28]. Intercellular communication mediated by EVs differs from that mediated by other molecules such as neurotransmitters, hormones, and cytokines in terms of the mode of release, mechanism of action, and uptake of the signal by the recipient cells. The first evidence is that EVs are released by almost all cell types; in contrast, neurotransmitters are released mainly by neurons while hormones by specialized endocrine cells. Cytokines, on the other hand, are released mainly by immune system cells but also by fibroblasts, epithelial and endothelial cells. The effect of neurotransmitters is fast and localized to other neuronal or muscle cells with specific postsynaptic receptors [29]. For hormones, the action is also mediated by specific receptors present on recipient cells, but it is usually a slower and more prolonged action. In fact, hormones, released into the bloodstream, also act on distant cells, whereas cytokines usually have a paracrine or autocrine action that modulates inflammatory and immune responses. EVs differ from the rest of the mediators in their ability to transfer bioactive molecular contents such as lipids, proteins, and mRNAs into their cargo by endocytosis, binding with receptors, or direct fusion with the membrane of recipient cells resulting in both short- and long-term effects [30].

In the last few years, a mounting amount of evidence has reported the existence of a bidirectional communication between the brain and other organs [31,32]. It has been suggested that this communication can be mediated by EVs that can cross the blood–brain barrier (BBB), transport different bioactive molecules (lipids, peptides, proteins, and nucleic acids), and can reach and target almost any organ [33]. For instance, recently, our group mechanically demonstrated the existence of a gut–brain axis mediated by EVs whose content seems to influence the central nervous system (CNS) function [34].

Over the years, several mechanisms by which EVs are able to pass the BBB have been discovered, including transcytosis, macropinocytosis, and endocytosis mediated by proteins such as clathrin and caveolin, but they can also cross it by direct translocation, for example under inflammatory conditions that increase barrier permeability [35]. The structural and molecular features that allow EVs to cross the BBB consist mainly in their small size, the lipid composition of the membrane that allows their fusion with those of endothelial cells, and the presence of surface proteins such as tetraspanins and integrins that can bind to endothelial cells’ receptors [36,37].

Here we summarize the existing literature data on possible brain–periphery communication that seems to be mediated by EVs, and, in particular, by EVs-delivered miRNAs. This mechanism has been clearly demonstrated in a few cases, but numerous findings indirectly suggest it could mediate this communication, especially in pathological conditions. Moreover, many data show that the altered expression of some peripheral tissue/organ-specific miRNAs may affect the CNS function and homeostasis and vice versa (Figure 1). Therefore, the knowledge of these processes could help understand and treat neurological and systemic diseases, for instance by modulating the delivery and the expression level of the involved miRNAs, likely by EVs-based therapies. In 2024, a meta-analysis was conducted to study the safety of using EV-based therapies to humans, and the results obtained showed that the therapies were safe and well tolerated with low frequency of adverse events [38].

## 2. Gut–Brain Axis

The gut–brain axis is a complex two-way communication system between the CNS and the gut. In this context, the gut microbiota (GM) plays a crucial role and the terminology has often been changed to the gut–brain–microbiota axis (GBMAx) [39].

The GM comprises symbiotic bacteria that inhabit the gut and play essential roles in maintaining gut balance and supporting host health. These bacteria participate in several critical functions, including metabolism of undigested nutrients, production of beneficial metabolites, protection from enteric pathogens, and immune system development [40]. Given their active function, recent findings have classified Eukaryotic organisms as meta-organisms, which function together with their microbiota [41].

Emerging research suggests that GM significantly influences brain health and disease. Gut dysbiosis, i.e., an imbalance or alteration of the gut microbiota composition, is believed to contribute to neurodegenerative and psychiatric conditions through various pathways including, oxidative stress, energy metabolism imbalance, and mitochondrial dysfunction [42,43]. For instance, in the context of Parkinson’s disease (PD), the pathological roles of α-synuclein seeds in gut–brain communication are increasingly recognized. Studies suggest that misfolded α-synuclein, a key pathological hallmark of PD, may originate in the gut and propagate to the brain via the vagus nerve. It is believed that this prion-like behavior of α-synuclein seeds is facilitated by intestinal dysbiosis and may be exacerbated by GM affecting the inflammatory state of the intestine and the integrity of the intestinal barrier. These processes contribute to the formation and propagation of α-synuclein aggregates, which are implicated in the pathogenesis of PD. Therefore, the gut–brain axis serves as a critical pathway in PD, with α-synuclein seeds acting as a mechanistic link between gut dysbiosis and neurodegeneration [44,45].

This paragraph examines the mechanisms linking GM to brain health, focusing on the correlation between GM dysbiosis and the onset of various neurologic disorders. In particular, we deepened the role played by EVs and by miRNAs in the GBMAx.

In the human meta-organism, EVs can be produced by both the host and the gut microbial community [46]. Human and microbial-derived vesicles act as modulators of various pathological conditions, including anxiety and mood disorders, such as major depressive disorder (MDD) [47], and in the development and progression of various types of diseases of the neurologic sphere, including neurodegenerative diseases such as Parkinson’s disease (PD), Alzheimer’s disease (AD), Multiple Sclerosis (MS) [48], as well as in neurodevelopmental disorders such as Autism Spectrum Disorder (ASD) [47]. miRNAs are important components of EVs’ cargo and their dysregulation has been associated with the GBMAx and the onset of various neurologic disorders [49,50].

Therefore, although no studies have definitively proven it, numerous findings suggest that the GBMAx may be mediated by miRNAs contained within EVs which could serve as a promising target/agent to modulate this system.

### 2.1. Gut–Brain–Microbiota Axis and EVs in Neurologic Disorders

Depression and anxiety are associated with impaired immune regulation and an inflammatory immune profile [51]. In this context, a role may be played by microvesicles (MVs) from commensal/probiotic origin that influence immune responses [48]. These MVs carrying microorganism-associated molecular patterns (MAMPs), such as lipopolysaccharide (LPS), peptidoglycans, and toxins, can cross tissue barriers including the BBB and interact with the immune receptors of glial cells (e.g., Toll-like receptors) to lead to the production of cytokines and neuroinflammatory mediators in the CNS, influencing brain function and behavior [52]. Also not to be underestimated is the fact that several bacteria, including lactobacilli, produce neurotransmitters (NTs), such as GABA, norepinephrine, serotonin, dopamine, and NTs precursors, that may be transported within MVs, reach the brain and regulate its activity and cognitive functions [52,53].

This opens new avenues for exploring therapeutic strategies targeting microbial interactions with the immune system and CNS to address mood and cognitive disorders.

### 2.2. Gut–Brain–Microbiota Axis and miRNAs in Neurologic Disorders

MDD is a psychiatric disorder characterized by significantly low and persistent mood. Some studies have shown a strong correlation between MDD, gut dysbiosis, and miRNA dysregulation. Chen et al. reported the presence of seven bacterial populations highly expressed in fecal samples from MDD patients (Anaerostipes, Bacteroides, Bifidobacterium, Clostridium, Collinsella, Dialister, and Roseburia) and correlated their presence with fecal miRNAs (miR-579-3p, miR-1246, miR-1276, miR-1976, miR-4488, and miR-3144-3p) involved in the regulation of specific depression pathways, suggesting the possible role of miRNAs in biological processes associated with MDD development [54].

Anxiety disorder is a chronic condition characterized by elevated levels of psychological stress with cognitive symptoms. A bioinformatic study demonstrated a positive correlation between gut dysbiosis and miR-206-3p expression in brain tissue. The consequent down-regulation of its target genes, Cited2 and STK39, led to the degeneration of mitochondria and synapses in the hippocampus, a brain region critical for mood regulation and cognitive function. This degeneration contributed to anxious behaviors, highlighting the importance of miR-206-3p in the pathogenesis of the pathological condition [55].

Fecal microbiota transplantation from generalized anxiety disorder (GAD) patients to germ-free mice induced anxiety-like behavior by altering the gut microbiota composition and the expression of some miRNAs in the prefrontal cortex of the recipient mice. It was found that the reduction of Akkermansia following fecal transplantation led to the upregulation of miR-488-3p expression and the downregulation of its target gene Creb1, resulting in anxiety-like behavior [56].

The GBMAx is also involved in the progression of neurodegenerative and neurodevelopmental illnesses including PD, AD, and ASD.

PD is a neurodegenerative disease characterized by the accumulation of α-synuclein protein in synaptic endings, leading to the loss of dopaminergic neurons and clinically presenting symptoms such as muscular rigidity [57]. Research findings suggest that gut dysbiosis participates in the etiopathogenesis of PD by promoting α-synuclein protein aggregation [58].

On the contrary, restoring intestinal microbial balance has shown promise in improving the clinical condition of PD patients. For instance, a study by Lee and colleagues analyzed the neuroprotective properties of Lactobacillus plantarum PS128 in a PD mouse model. The treatment with PS128 reduced the level of miR-155-5p and restored the presence of three bacterial species: Bifidobacterium, Ruminclostridium Bacteroides, and Alistipes, which were statistically correlated with improvement in rotenone-induced motor deficits [59].

AD is another neurodegenerative disorder associated with the accumulation of β-amyloid protein, leading to dementia [60]. The composition of the gut microbiota and dysbiosis strongly impact the onset and progression of AD. It has been shown that the infection of human neuronal-glial (HNG) cells with the neurotropic herpes simplex virus-1 (HSV-1) and the gastrointestinal tract abundant Gram-negative bacillus Bacteroides fragilis induced the upregulation of the NF-κB-related pro-inflammatory miRNAs, such as miR-146a, resulting in neuroinflammation and neurodegeneration contributing to AD [61,62,63].

ASD is a neurodevelopmental disorder primarily affecting language and communication. Emerging studies link ASD to an imbalance in the gut microbiota and altered miRNA levels [64,65].

A study showed that the expression levels of some circulating miRNAs associated with gastrointestinal disorders (GID), including miR-320-5p, miR-31-5p and miR-223-5p, were significantly higher in children with ASD compared to controls, suggesting its contribution to disease development [66].

The spectrum of neurological diseases involving the gut microbiota also extends to MS. A study conducted on fecal samples from patients with relapsing-remitting multiple sclerosis (RRMS), identified different expressions of certain miRNAs, including miR-146a, compared to control samples [67].

Fecal miRNAs, most of which are present in fecal EVs derived by both the intestinal microbiota and the host, can be considered as circulating miRNAs and have emerged as a powerful tool for disease diagnosis and the development of miRNA-based therapeutic strategies, especially for intestinal-related diseases such as colorectal cancer and celiac disease [68]. The findings reported above showed that altered fecal miRNA expression also correlates with brain dysfunctions, indirectly suggesting that these miRNAs may spread through the body and act at a systemic level.

These findings indicate potential therapeutic targets, although further studies are necessary to confirm the efficacy of miRNAs as treatments. In this context, there are numerous ongoing investigations into using EVs-delivered miRNAs as therapeutic agents. Most of these clinical trials and studies primarily focus on cancer and neurodegenerative diseases [69,70]. However, most efforts are still in the experimental or early clinical trial stages, with challenges related to safety, delivery, and efficacy remaining to be addressed. Therefore, the knowledge of these mechanisms could not only allow a greater understanding of the pathophysiological mechanisms underlying mental disorders and neurodegenerative diseases, but also the development of potential therapeutic strategies to maintain or restore deregulated mechanisms.

## 3. Lung–Brain Axis

The concept of the brain–lung axis involves bidirectional communication between the lung and the CNS that operates through several mechanisms, such as direct translocation of microorganisms, microbiota-mediated effects on nerves, the hypothalamus–pituitary–adrenal axis (HPA), metabolic changes and immune pathways [71,72]. Recent studies have shown that another key mechanism of brain–lung communication involves EVs, both in physiological and pathological conditions [73,74]. For instance, it has been observed that higher circulating EVs from adenocarcinoma cells in plasma samples from lung cancer patients were positively associated with a higher risk of stroke and more extensive brain damage in these patients [75].

The COVID-19 pandemic has also provided insights into the interaction between the lung and the brain, with SARS-CoV-2 infection associated with neural manifestations such as ischemic stroke and chronic fatigue syndrome [76]. Furthermore, studies have shown that elevated levels of inflammatory factors were found in both the brain and the lung of mice with traumatic brain injury (TBI) [77]. Notably, the brain-derived EVs play a crucial role in transporting inflammatory proteins to the lung, leading to lung inflammation. On the contrary, the targeted blockade of brain-derived EVs uptake in the lung alleviated lung inflammation in TBI mice [77].

Therefore, the interaction between the lungs and the brain is evident through compelling indications that injuries and inflammation in one organ can impact the susceptibility of the other, primarily through the circulation of immune cells and inflammatory mediators [78,79]. For instance, pneumonia can worsen brain injury after a stroke, and some stroke patients may develop acute lung injury, pneumonia, or bronchitis during hospitalization [80]. Additionally, chronic obstructive pulmonary disease (COPD) is associated with cerebrovascular disease, increasing the risk of ischemic and hemorrhagic stroke [81].

Existing studies have found that miRNAs also function through the intercellular transfer of EVs, which play a crucial role in protecting and sorting miRNAs [82]. Since the EVs are involved in communication between peripheral organs and the brain, this evidence suggests a primary role for miRNAs in the lung–brain axis through established pathways as well as potential new ones. The understanding of the lung–brain axis is still emerging and limited available data point towards epigenetic regulation, including post-transcriptional mediation of miRNAs, as a significant mechanism influencing the development of numerous phenotypes and endotypes of chronic respiratory diseases, such as asthma. Recent evidence has indicated the potential of miRNAs, such as miR-338-3p, miR-1246, and miR5100, to serve as biomarkers and therapeutic targets in this context [83,84].

Recent studies have uncovered correlations between lung cancer and brain metastases (BM) formation, highlighting the role played by cancer cells-derived EVs cargo, especially miRNAs [85,86,87]. BM are a common complication of both small-cell and non-small-cell lung cancer (NSCLC) and different miRNAs have been identified as playing a role in developing these metastases [88]. miR-21 is involved in several processes associated with malignancies and is overexpressed in different cancers. On the contrary, its inhibition has been demonstrated to diminish tumor cell proliferation, migration and invasion in a range of cancer types, including pancreatic, colorectal, gastric, pulmonary and non-small cell lung cancers [89]. NSCLC patients with BM exhibited elevated levels of miR-21 compared to those without metastases, indicating that high miR-21 expression may be associated with BM progression [90]. According to this result, Zhu and colleagues observed that low serum levels of miR-21 were associated with a longer survival of NSCLC patients with BM [91].

Zhao et al. found that miR-145 was downregulated in BM samples compared to primary lung adenocarcinoma samples, and its upregulation reduced the proliferation of lung adenocarcinoma cells, likely by targeting some proliferative pathways. The authors suggested this miRNA plays a pivotal role in the proliferation and colonization of lung adenocarcinoma cells in brain metastatic sites, and thus could be a valuable target in cancer therapy [92].

It has been demonstrated that miR-217 suppressed metastatic progression and BM formation from NSCLC by targeting SIRT1 and activating the P53/KAI1 signaling pathway. Therefore, the regulation of the miR-217/SIRT1/P53/KAI1 pathway might provide a promising therapy to prevent the formation of BM from NSCLC [93].

A direct involvement of EVs–miRNA in promoting BM development from NSCLC has been recently reported by Jin and colleagues [87]. The authors demonstrated that serum NSCLC-derived EVs-associated miR-374a-5p promoted premetastatic niche formation by reducing BBB permeability and facilitating leptomeningeal metastases (LM) formation, which reduced lung cancer patients’ survival [87]. Therefore, this miRNA could serve as a blood biomarker for the diagnosis and prognosis of NSCLC-derived LM.

In summary, communication in the brain–lung axis not only involves the vagus nerve, endocrine and immune mechanisms but also sees an active role of miRNAs. This could provide new insights into the development of diseases related to the brain–lung axis and new insights about possible biomarkers and therapeutic targets.

Despite the considerable research conducted in recent years, the understanding of the lung–brain axis remains incomplete. Further research is required to elucidate the specific mechanisms by which miRNAs influence this complex interaction. For instance, some studies have sought to elucidate the roles of specific miRNAs in facilitating the spread of lung cancer cells to the brain and the formation of BM [94]. These investigations often begin with in vitro analyses to define the transcriptome of brain metastatic lesions and to identify miRNAs that are significantly overexpressed or downregulated. Following this, functional studies are performed to evaluate the biological consequences of modulating the expression of these miRNAs. By reversing their expression levels, either through suppression or enhancement, researchers aim to clarify their influence on cell behaviors such as migration, invasion, proliferation, the disruption of the blood–brain barrier, and so on. These combined approaches could help clarify the molecular mechanisms underlying BM formation and identify potential therapeutic targets for intervention.

## 4. Heart–Brain Axis

The principle that EVs cargo mediate the intercellular communication and can act as gene regulators is also applicable to the cardiovascular system.

Cardiac cells-derived EVs contain about 1500 different mRNAs, 340 DNA sequences, and over 40 specific proteins, which are involved in a wide range of biological processes [95,96]. In addition, heart-derived EVs are implicated in communications with other organs and tissues, particularly with the CNS, adipose tissue, and skeletal muscle [95]. In fact, cardiac cells-derived EVs can easily cross the BBB to reach the CNS.

EVs are produced also by pericardial adipocytes and, remarkably, by Adipose Tissue-Derived Stem Cells (ADSCs). The latter are involved in neuronal regeneration, axonal growth, proliferation, migration and myelination of Schwann cells processes [95]. Moreover, cardiovascular diseases can be responsible for neurodegenerative phenomena and cognitive disorders known as “cardiogenic dementia”, that derive both from the reduction of blood flow to the brain and from the release of neuroactive humoral molecules into circulation [97].

Numerous findings suggest that these multiple heart–brain communications may be mediated by heart-derived EVs cargo, particularly by the contained miRNAs. In healthy heart, the most expressed miRNAs are miR-1, miR-133, miR-126-3p, miR-143, miR-145, miR-30c and miR-22 [98].

miR-1 is a cardio-specific miRNA involved in cardiac development, myocardial infarction (MI), arrhythmogenesis and heart failure [95,98]. In a mouse MI model, it has been observed that the overexpression of miR-1 transported by EVs from infarcted hearts in the hippocampus, induced local TPPP/p25 protein downregulation, resulting in neuronal microtubule damage [99]. Therefore, the obtained results suggest the use of this miRNA as a useful diagnostic marker, as well as a possible therapeutic target in clinical practice for the treatment of cardiovascular disorders [99].

In a rodent model of chronic heart failure (CHF), it was demonstrated that cardiomyocytes-derived EVs may act on the CNS, especially in the rostral ventrolateral medulla (RVLM) area, contributing to oxidative stress and sympathetic excitation via a miRNA-mediated mechanism [99]. Here, the miRNAs contained inside heart-derived EVs (mainly miR-27a) induced the inhibition of the Nrf2 pathway, resulting in sympathetic hyperactivity, baroreflex impairment, increased afterload and consequent cardiac hypertrophy, as well as increased oxidative stress. All these effects, resulted in a vicious circle in which cardiac pump dysfunction causes an increase in the aforementioned miRNAs levels, generates neuroinflammation, leads to sympathetic hyperstimulation, and further complicates heart failure [100].

It has been observed that up to 11–18% of stroke patients have symptomatic heart failure, ischemic-like electrocardiographic changes, cardiac structural changes, and arrhythmias, suggesting a clear bidirectional communication between the heart and the CNS, that could be mediated by EVs content [101]. Accordingly, some similarities between miRNAs contained in EVs released during cardiac or cerebral ischemia have been observed, including a reduction in miR-340 and miR-424 levels and an increase in miR-29b levels [102]. Physiologically, miR-424 promotes angiogenesis, and miR-340 plays a protective role against atherosclerotic phenomena by interacting with APO-B and LRP1 proteins and reducing the inflammatory status. miR-29b has a protective role against both acute MI (AMI) and stroke ischemic damage by negatively regulating p53-associated apoptosis [102].

Other relevant miRNAs are those derived from mesenchymal stem cells (MSCs). In an AMI mouse model, human MSCs-derived exosomes can reduce the area of ischemic necrosis and enhance myocardial cells viability, by increasing ATP and nicotinamide adenine dinucleotide levels, reducing ROS production and oxidative stress, as well as by activating the PI3K/Akt pro-survival signaling pathway in cardiac tissue [103]. These effects were likely mediated by the miRNAs enriched in these MSCs-derived EVs. In fact, different studies reported that exosomes originating from mouse/rat MSCs contained a large amount of miR-22 and miR-19a that were able to exert an antiapoptotic effect and promote cardiomyocytes survival by down-regulating their targets, including methyl-CPG binding protein 2 and PTEN [104,105].

miR-17–92 cluster has been found inside MSCs-derived EVs and shown to regulate cardiomyocyte proliferation in the neonatal heart, ischemic damage repair in the adult heart, as well as post-stroke neural progenitor cells proliferation [106,107,108]. This protective effect depends on miR-17–92 cluster-dependent post-transcriptional regulation. In fact, both in clinical practice and animal models, miR-17-92 cluster protects both heart and brain from ischemic damage by promoting the PI3K/AKT, and MAPK/ERK pro-survival pathways [108].

Thus, a bidirectional communication between heart and brain exists, especially in pathological conditions, and numerous findings suggest it could be mediated by EVs-delivered miRNAs. However, it has not been defined yet whether all these molecules can be used as disease biomarkers or as therapeutic targets in clinical practice, and further research are needed.

## 5. Skeletal Muscle–Brain Axis

Skeletal muscle is one of the human body’s most dynamic and plastic tissues and contributes to multiple functions. From a mechanical point of view, the muscle’s main function is converting chemical energy into mechanics to generate strength and power, maintain posture, and produce movement. From a metabolic point of view, it acts as a deposit for important substrates, such as amino acids and carbohydrates, which are involved in the production of the heat necessary to maintain the internal body temperature at the basis of physical activity [109]. Over the past two decades, it has become apparent that skeletal muscle also functions as an endocrine organ, capable of producing and secreting hundreds of bioactive molecules, which exert their effects in an autocrine, paracrine, or endocrine manner [110]. Among these molecules, the best known are myokines, small peptides synthesized and released by myocytes in muscle tissue in response to external lesions or stimuli. Myokines are implicated both in the autocrine regulation of muscle metabolism and in the paracrine and endocrine regulation of other tissues and organs, such as adipose tissue, liver, and brain [111].

The relationship between muscle and the nervous system is widely known by the term of “muscle–brain axis” particularly in the response to exercise [112,113,114]. It has been observed that physical exercise reduces age-related brain atrophy [115], the risk of onset and progression of neurodegenerative diseases [116], oxidative stress and neuroinflammation [117], as well as improves brain metabolic functions and mitochondrial biogenesis [118]. In such a relationship, the role of myokines (e.g., insulin-like growth factor, IGF-1; acidic and cysteine-rich secreted protein, SPARC; irisin; lactate; interleukin 6, IL-6) is well studied as messengers for communication between the skeletal muscle and the brain [119]. Currently, miRNAs circulating and contained within EVs, as mediators of intercellular and inter-organ communication have also been recently described in the “muscle–brain axis” context [120], following damage, pathological conditions, and/or stimuli (e.g., exercise), underlying several molecular mechanisms in the brain. Once released by the muscle into the bloodstream, these molecules can cross the BBB and deeply affect brain functions such as neurogenesis, cognitive functions, neuronal plasticity, and neuroinflammation [121,122].

Of note, skeletal muscle expresses a series of tissue-specific miRNAs called “myomiR”, which include miR-1, miR-133a/b, miR-206, miR-208a/b and miR-499a [123]. The expression of these molecules seems to be both modulated by physical exercise [124], and altered in the presence of different pathologies, suggesting their potential diagnostic and prognostic role [125]. For instance, miR-1 and miR-133a are upregulated in response to endurance and resistance exercise, facilitating myogenesis and adaptation to increased mechanical load [126,127]. Similarly, miR-206, a myomiR involved in muscle repair, is elevated following muscle damage induced by exercise, suggesting its role in promoting regeneration and communication with other tissues, including the nervous system [124,128]. Moreover, the release of these myomiRs into the bloodstream during exercise further supports their role in inter-organ communication. These circulating myomiRs can reach the brain, where they may influence neuroplasticity, modulate inflammation, and counteract oxidative stress, contributing to the neuroprotective effects of regular physical activity [129].

One of the sights that arouses particular interest in crosstalk between muscle and brain is miR-206. Several studies suggest that miR-206, including circulating miR-206, regulates the expression of brain-derived neurotrophic factor (BDNF) [128,130,131], a member of the neurotrophin growth factor family that plays a crucial role in the development of CNS, supporting the survival of existing neurons and stimulating neurogenesis. In physiological conditions, the level of miR-206, being specific to skeletal muscle, is not detectable or is too low to be detected in the brain. Conversely, it is highly expressed in neurological diseases, including AD, suggesting that miR-206 could participate in the progression of the disease by inhibiting the expression of BDNF [128]. Serum exosomal miR-206 levels were also found to be altered in patients with brain trauma. Specifically, subjects with elevated exosomal miR-206 levels, induced by the reduction of BDNF, showed poor prognosis [130]. In addition, BDNF also plays a critical role in pathological nociceptive processes, as in neuropathic pain [132]. Similarly, miR-1 negatively regulates BDNF expression, directly targeting the untranslated 3′ region (UTR) of its mRNA [133]. In turn, inhibition of BDNF appears to act on downstream targets associated with pain, including Heat Shock Protein 60 (Hsp60) and Connexin 43 (Cx43) [134]. Considering this, the lack of Hsp60 and Cx43, which play essential functions in somatosensation, can contribute to the onset of neuropathic pain mediated by miRNAs. In addition to knowing about the existence of a muscle-brain axis, this information suggests that exosomal miRNAs released by skeletal muscle could serve as biomarkers for diseases of the nervous system, offering a method to assess the severity and prognosis of specific neuronal diseases.

Overall, skeletal muscle plays a fundamental role in systemic physiology. Given its extensive mass, it is not surprising that changes in myomiR regulation are associated with pathological muscle conditions (e.g., sarcopenia, muscular dystrophy, etc.), which are responsible for influencing miRNAs profiles in plasma and serum. However, these changes not only reflect the alteration of the muscular state but can have important implications for the health of the nervous system, confirming a close interconnection between muscle health and neurological function. For instance, murine models of muscle atrophy showed elevated levels of plasma exosomal miR-29b-3p, which if crossing the BBB could reach the nervous system and affect neuron functions [135]. This study confirmed that C2C12 muscle cells-derived exosomal miR-29b-3p can be up-taken by the neuronal cells SH-SY5Y, inducing the downregulation of neuronal genes, such as BCL-2 and LAMC1, which are responsible for the process of neuronal differentiation [135]. Another myomiR is miR-486, which is normally involved in muscle functions [136] but is also overexpressed in the presence of spinal cord injury. Recently, a possible relationship has been described between miR-486 and the expression of NeuroD6 [137], a protein responsible for neuronal differentiation and survival, through the expression of glutathione peroxidase 3 (GPX3) and thioredoxin-like 1 protein (TXNL1) [138].

miR-499 is responsible for the proliferation and differentiation of myoblasts [139]. High serum levels of miR-499 have been reported in patients with TBI [140], and during the development of bipolar disorder (BD) [141]. In the latter case, miR-499 has been shown to target the CACNB2 gene, coding for an L-type voltage-dependent calcium channel-regulating subunit (LVGCC), contributing to dendritic impairments, deregulated calcium homeostasis, and neurocognitive dysfunction [141]. Since miR-499 is specific to the skeletal and cardiac muscles, consequently expressed at low levels in the CNS, it was possible to hypothesize that miR-499 intervenes by modulating the neuronal activity or inflammatory responses.

miR-133a is known to contribute to myoblast proliferation and differentiation [126,127]. Moreover, circulating myo-miR-133a-3p has been found overexpressed in several pathological conditions, characterized by muscle weakness and progressive muscle wasting [125,142,143]. For instance, circulating myo-miR-133a-3p levels increased in individuals affected by Cushing’s syndrome compared to healthy subjects, and positively correlated with urinary free cortisol level, suggesting it as a promising biomarker to assess disease severity [144].

In summary, the muscle–brain axis is emerging as a critical pathway of inter-organ communication, through which miRNAs, produced by muscle during exercise or in response to pathological conditions, can affect brain function. The increase of the specific myomiR in the CNS in some pathological conditions may indicate compensatory responses or communication mechanisms aiming to maintain homeostasis and neuronal function. In addition, the ability of some myomiR to act as biomarkers for neurodegenerative conditions or brain lesions opens new perspectives for the early diagnosis and monitoring of the prognosis of these pathologies. Overall, further research is needed, not only to decipher the complex molecular mechanisms that regulate this relationship but also to pave the way for new strategies for the treatment of neurodegenerative diseases and the improvement of cognitive function through targeted interventions on miRNAs.

## 6. Adipose Tissue–Brain Axis

The classical functions of adipose tissue (AT) are energy storage, regulation of metabolism through the release of fatty acids into the circulation, thermal isolation, and mechanical protection. In recent years, AT has been studied and investigated further and is now considered among the organs of our body [145]. Following its identification as an active endocrine organ, studies have been conducted to identify the various hormones and cytokines it produces and releases, which regulate physiological functions such as satiety and glucose and lipid metabolism or pathological conditions such as diabetes, obesity, and inflammation [146].

Furthermore, AT communicates and interacts with other systems and apparatus including the nervous system. Therefore, an AT–brain axis is hypothesized to exist [147,148]. A well-known example of the presence of an AT–brain axis is the secretion of the hormone leptin from adipocytes (AT cells). Leptin is involved in the regulation of appetite and acts on brain hypothalamic receptors with the final effect of satiety and reduction of food intake [149]. The communication between these two organs is bidirectional since AT is innervated by the sympathetic nervous system, and in addition, the brain releases neuropeptides (e.g., corticotropin-releasing hormone CRH and thyrotropin-releasing hormone TRH) that also influence AT metabolism [150,151].

Communication between the AT and the brain can also occur through the secretion of EVs, which can modulate brain processes such as neurogenesis and synaptic plasticity, as well as energy metabolism and the neuroinflammatory response, as they cross the BBB [152,153,154]. In fact, patients with type 2 diabetes (T2D) and obesity-related insulin resistance often also have an increased risk of cognitive impairment [155]. Treatment of primary neurons with medium conditioned with the AT from High Fat Diet (HFD)-fed mice induced synaptic loss and consequently decreased synaptic plasticity leading to learning and memory problems underlying many neurodegenerative diseases [156]. Pathways regulated by adipose-derived EVs in overweight patients include stimulation of monocyte differentiation and macrophage activation, but also upregulation of tumor necrosis factor-α (TNF-α) and interleukin-6 (IL-6), resulting in inflammation and dysregulation of body homeostasis [157]. The presence of inflammatory miRNAs such as miR-155, miR-21, and miR-425, and pro-inflammatory cytokines within EVs released from AT under conditions of obesity may contribute to the neuroinflammation that underlies many brain diseases [158,159]. miR-29a was found overexpressed in AT macrophages-derived exosomes of obese individuals and associated with obesity-induced tissue inflammation and insulin resistance [160]. This typical AT-associated miRNA has been shown to play a role in neurodegeneration as its levels increase in the cerebral spinal fluid (CSF) of patients with AD [161]. Another example of the axis between brain and AT is the altered expression of miR-33 which is involved in both cholesterol homeostasis and increased Aβ levels in the brain, resulting in protein aggregation and synapse destruction [162]. A 2022 study found that miR-9-3p is upregulated in both the hippocampus and AT-derived vesicles of HFD-fed mice, and the same trend of results was present in adipose-derived vesicles of patients with diabetes [156]. The authors demonstrated that miR-9-3p-mediated downregulation of BDNF is responsible for synaptic damage and consequent cognitive decline in HFD-fed mice [156].

MALAT1, miR-181b and miR-144 were found increased within EVs released from AT of obese mice [163]. The administration of these EVs to lean mice increased appetite and weight, by regulating the rapamycin (mTOR) signaling pathway in Pro-opiomelanocortin (POMC) neurons, suggesting their involvement in the interaction between adipocytes and hypothalamic neurons [163].

It was seen that miR-200a/b, miR-429 and miR-488 were correlated with leptin levels in obese mice, suggesting a key role in the AT–brain axis, as their expression was also altered in the hypothalamus of HFD mice [164,165,166].

Considering these findings, it is imperative to further investigate the bidirectional communication between AT and the brain. Indeed, this research would have the potential to uncover new therapeutic strategies that could benefit both metabolic diseases, such as type 2 diabetes, and neurological diseases associated with metabolic dysfunction, such as AD. This may pave the way for more effective treatments that target the mechanisms underlying these interconnected conditions.

## 7. Kidney–Brain Axis

The brain and kidney communicate under physiological conditions such as through the release of hormones like vasopressin and aldosterone but also during pathological conditions. The latter include, for example, chronic kidney disease (CKD) where patients can also develop neurological complications [167]. Communication between the two organs is always bidirectional, and in fact cerebrovascular disease can result in renal dysfunction [168].

EVs play an active role in the crosstalk between the two organs and have gained attention in recent years to analyze the pathological mechanisms involved in, for example, acute kidney injury (AKI), CKD, or diabetic nephropathy [169]. For instance, mice with AKI and CKD were seen to secrete high levels of EVs from the kidneys and in the urine compared with controls, and their contents contained high levels of inflammatory cytokines [170].

A study of elderly patients with end-stage renal disease (ESRD) showed altered levels of circulating angiogenic miRNAs and specifically higher levels of miR-29a and lower levels of miR-223, miR-27a, and miR-326 that were associated with worse cognitive function [171].

In a pilot study, sequencing of brain- and kidney-derived EVs was carried out in mice deficient in an enzyme implicated in both the pathogenesis of neurodegenerative diseases and diabetic nephropathy, namely l-isoaspartyl methyltransferase (PIMT), and it was seen that vesicular miR-34a was overexpressed in both kidney- and brain-derived EVs [172].

Furthermore, in patients with CKD, alterations in the cerebral circulation cause ischemic strokes and behavioral disorders, and it was seen that the expression of miR-17 and miR-126 in the endothelial cells of cerebral arterioles of mice with CDK compared with those whose WT is deregulated, suggesting their use as novel biological markers of brain disorders in CKD patients [173]. The investigation for common biomarkers is still immature but may lead to useful new tools in the future to prevent neurological complications following kidney disease and vice versa by liquid biopsy.

In addition, the onset of CKD has been shown to be related to the aging process of the kidney and thus to the phenomenon of cellular senescence [174]. EVs are part of the senescence-associated secretory phenotypes (SASPs), and therefore it would be useful to properly study their vesicular content, including miRNAs or their use as drug carriers for anti-aging therapies in the kidney with decreased adverse effects in other organs such as the brain.

## 8. Immune–Brain Axis

Over the past few years, numerous findings have supported the concept of an immune–brain axis, which refers to the dynamic bidirectional communication between the immune system (IS) and the CNS that crosstalk extensively via multiple pathways to ensure the health and the homeostasis of the entire organism [175].

One of the pathways of this bidirectional communication is that mediated by EVs delivering cytokines, chemokines and other inflammatory factors promoting neuroinflammation [176,177].

Also, in this case, EVs-delivered miRNAs seem to play a key role. For instance, it has been observed that the exposure of neurons, various glial cells, and brain endothelial cells to inflammatory stimuli upregulated miR-146a expression which targets mRNAs encoding for proteins involved in cellular energy metabolism, leading to the loss of mitochondrial integrity and function. Moreover, an elevated miR-146a level has also been found in the secreted EVs, suggesting the propagation of a signal promoting cellular bioenergetics deficit and reinforcing neuroinflammatory response [178].

miR-155 is a pro-inflammatory mediator of the CNS and the modulation of its expression could be used as a therapeutic strategy in the treatment of pathological neuroinflammation such as MS, AD and other neuroinflammatory disorders [179]. For instance, it has been reported that miR-155 is a regulator of α-synuclein and induced an inflammatory response in PD [180]. Moreover, a high level of miR-155 was detected in EVs from neurons of PD patients, likely promoting the propagation of inflammatory responses [181].

miR-124 is one of the most abundant miRNAs in the mammalian CNS, where it plays an anti-inflammatory role by promoting the transformation of microglia cells from the pro-inflammatory M1 type to the anti-inflammatory M2 type and maintaining them in a quiescent state [182]. Consequently, altered expression of miR-124 has been associated with multiple neuroinflammatory disorders such as PD and AD [183,184], suggesting it as valuable therapeutic agent. In agreement, a recent study demonstrated the possibility of engineering EVs to deliver miR-124 into the CNS and alleviate cocaine-mediated microglial activation [185].

It has been found that the miR-21-5p level increased in neurons and microglia after traumatic brain injury. Neurons-derived exosomes containing miR-21-5p were phagocytized by microglia and induced M1 microglia polarization, aggravating the release of neuroinflammatory cytokines, inhibiting the neurite outgrowth, and promoting neurons apoptosis [186,187].

Beyond EVs released from neuronal and glial cells, other cells of the IS may induce neuroinflammation via EVs-delivered miRNAs. For instance, recently, Hu and colleagues reported that exosomal miR-409-3p released by activated mast cells enhance microglial migration, activation, and neuroinflammation by inducing the NF-κB pathway, suggesting exosomal miRNAs as promising therapeutic targets [188].

Collectively, although the immune–brain axis is still an evolving field, growing evidence strongly supports its existence and the reported data suggest a possible involvement of EVs and EVs cargo as mediators of this bidirectional communication.

## 9. Conclusions

The delivery of miRNAs through EVs is among the intercellular communication processes that have been increasingly emerging over the years. Currently, there are several methods for the isolation of EVs (such as ultracentrifugation, size-exclusion chromatography, precipitation methods, commercial kits and specific immunoaffinity techniques), and their characterization is based on both the analysis of their cargo and the use of instruments that assess their morphology and size, such as Nanoparticle tracking analysis (NTA), Atomic Force Microscopy (AFM), and Transmission electron microscopy (TEM) [189]. Despite the advances in these technologies, however, it will be necessary optimize and standardize these methods to obtain reproducible data among different studies. In this review we wanted to focus on their role as part of the possible axes between the brain and peripheral organs such as the gut, lung, heart, skeletal muscle, adipose tissue, kidney and the immune system. Reported in vitro and in vivo studies confirm their potential role both as novel biomarkers and as possible therapeutic targets primarily for the treatment of neurodegenerative and systemic diseases. In this context, it is therefore necessary to continue to research the role of EVs-delivered miRNAs in physiological processes and in the development of pathological conditions. Understanding these mechanisms would, in fact, make them excellent biomarker candidates for early diagnosis and monitoring of disease prognosis such as the development of brain metastasis following lung cancer. In conclusion, we wanted to emphasize the importance of EVs and their cargo in the future landscape of medicine, which will be oriented toward a comprehensive evaluation of the possible axes of communication between the various organs and not to the treatment of a single diseased organ, and which also considers meta-organs such as the human-hosted microbiota.

## Figures and Tables

**Figure 1 biology-13-01056-f001:**
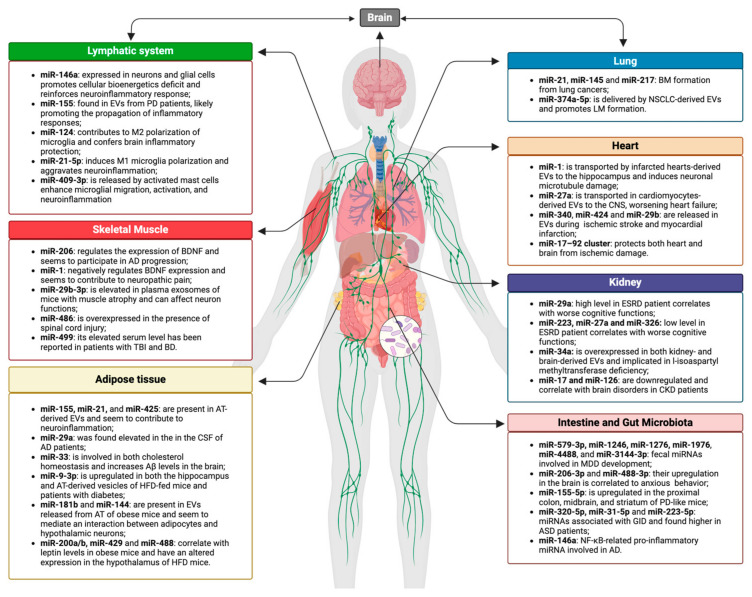
**miRNAs involved in brain–periphery axes.** The figure lists the miRNAs that have been discussed in the text and that seem to mediate a direct/indirect communication between the CNS and peripheral organs/tissues (i.e., lungs, heart, gut and gut microbiota; skeletal muscle, adipose tissue, kidney and immune system). Some of these miRNAs are transported by organs/tissues derived EVs directly to the brain, affecting its functions and homeostasis (the image was created with BioRender.com online software: https://www.biorender.com Accessed on 8 November 2024).

**Table 1 biology-13-01056-t001:** Mechanisms of target genes expression regulation by miRNAs.

Target Region	Effect on Target Genes Expression	References
Target-mRNAs 3′ UTR	Target-mRNAs destabilization and translation inhibition	[2]
Target-mRNAs 5′ UTR	Target-mRNAs destabilization and translation inhibition Enhancement of target mRNAs stability and translation	[3,4]
Target-mRNAs coding sequence	Target-mRNAs translation inhibition	[5,6]
Target-genes promoters	Target-genes transcription activation	[7]

**Abbreviations**: mRNA, messenger RNA; UTR, untranslated region.

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
