# Peer review of "Brain–Periphery Axes: The Potential Role of Extracellular Vesicles-Delivered miRNAs"

_biology, 2024, doi:10.3390/biology13121056_

Round 1
Reviewer 1 Report
Comments and Suggestions for Authors
Overall this is a nicely organized review with good information that contributes to the field. At this onset, the functional mechanism of miRNAs is not discussed. Perhaps rooting the paper in basic miRNA biology would improve understanding and prevent generalized statements about miRNAs primarily functioning in EVs as this is not a generally held belief and there was not primary literature cited to back up the claim. The gut-brain axis section as it is a bit light on EV-related research which strays from the title of the review.
Additionally, it would be good to highlight miRNAs in the graphical abstract since that is the focus of the review. It would be helpful to include a list of some notable EV miRNAs mentioned in each section – perhaps miRNAs listed next to specific organs. Currently, the graphical abstract connects peripheral organs and the brain via EV cargo in circulation (some examples go beyond circulation).
Line 61, reference 9: Cataracts are the only condition being referenced.
Line 61-63 could use rephrasing as they are promising, but not common use or validated in a clinical setting.
Line 66, “extracellular fluids”: Biofluids? Unclear what exactly is being referred to. could say "(found in serum..)". Serum, plasma, etc., are not extracellular fluids, but they have an extracellular environment along with other things.
Line 73, references 14-18: References are focused on cancer/tumors.
Gut-Brain Axis and miRNAs in neurological disorders: References are not extracellular vesicle-related.
Line 99 The GBMAx term is only used once and it is unclear how it is different from a different term used in line 113 or the multiple times after that gut-brain axis is spelled out in line 123 and line 124.
Section 2.2 does not reference miRNAs that are in circulation and it is unclear how it is proposed that miRNAs present in the feces are entering circulation and impacting the brain.
Line 195, “further studies are necessary to confirm the efficacy of miRNAs as treatments”: Is this a stagnant area, or are studies on the way? Are there studies using EVs themselves as miRNA transport as a treatment?
Lines 227-228: This statement should be rephrased. miRNAs primarily function within their tissue of origin. Additionally some work indicates they also function in intercellular communication through EVs. This statement is not citing a primary literature article, so hard to say the evidence that backs it up.
Line 249 and 26: NSCLC - typo
Line 274: How would identifying newer miRNAs lead to further understanding of the understanding of the lung-brain axis? Are there any specific pathways of interest? What would these studies look like?
Line 369: word choice should be modified to inter-organ
Line 440-443: How does exercise relate to these changes in 'myomiRs" in skeletal muscle.
Define POMC in line 490.
Author Response
Reviewer#1 Comment #1
Overall this is a nicely organized review with good information that contributes to the field. At this onset, the functional mechanism of miRNAs is not discussed. Perhaps rooting the paper in basic miRNA biology would improve understanding and prevent generalized statements about miRNAs primarily functioning in EVs as this is not a generally held belief and there was not primary literature cited to back up the claim. The gut-brain axis section as it is a bit light on EV-related research which strays from the title of the review.
Authors’ reply
We thank the Reviewer for this comment. We added a brief description of miRNAs biogenesis, and briefly discussed the different modalities of action already summarized in Table 1 (new lines 51-59 and 61-65). Moreover, we added a statement, with proper citations (new lines 77-79) to highlight the abundance of tissue-specific miRNAs compared to the circulating ones.
Regarding the paragraph on the gut-brain axis, since there is very little literature about the direct involvement of EVs-delivered miRNAs, we decided to add some information about the EVs mediating this communication, indirectly suggesting the participation of the contained miRNAs.
Comment#2
Additionally, it would be good to highlight miRNAs in the graphical abstract since that is the focus of the review. It would be helpful to include a list of some notable EV miRNAs mentioned in each section – perhaps miRNAs listed next to specific organs. Currently, the graphical abstract connects peripheral organs and the brain via EV cargo in circulation (some examples go beyond circulation)
Author’s reply
We thank the Reviewer. We modified the graphical abstract according to the suggestion.
Additional areas to be addressed:
Comment #3
Line 61, reference 9: Cataracts are the only condition being referenced.
Author’s reply
We thank the reviewer for this suggestion. We cited other works reporting as miRNAs dysregulated expression pattern may be correlated to different pathologies (new references 11-14).
Comment #4
Line 61-63 could use rephrasing as they are promising, but not common use or validated in a clinical setting.
Authors’ reply
We thank the Reviewer for this feedback. We reformulated the sentence (new lines 74-76).
Comment #5
Line 66, “extracellular fluids”: Biofluids? Unclear what exactly is being referred to. could say "(found in serum..)". Serum, plasma, etc., are not extracellular fluids, but they have an extracellular environment along with other things.
Authors’ reply
We thank the Reviewer for this comment. We reformulated the sentence (new line 79).
Comment #6
Line 73, references 14-18: References are focused on cancer/tumors.
Authors’ reply
We thank the Reviewer for this comment. We added some other pertinent citations (new references 26-28)
Comment #7
Gut-Brain Axis and miRNAs in neurological disorders: References are not extracellular vesicle-related.
Authors’ replay
We thank the reviewer for this comment. We are aware that the cited references do not talk of EVs-delivered miRNAs mediating the gut-brain axis. However, as declared in the introduction (new lines 127-131) communication between the brain and peripheral tissues/organs mediated by EVs-delivered miRNAs has been clearly proved in few cases. For the others, the current literature only reports some findings suggesting this mechanism.
To highlight this concept we added some lines in the gut-brain axis paragraph (new lines 183-185).
Comment #8
Line 99 The GBMAx term is only used once and it is unclear how it is different from a different term used in line 113 or the multiple times after that gut-brain axis is spelled out in line 123 and line 124.
Authors’ reply
We thank the Reviewer for this comment. We changed the text, using the same acronym, i.e. GBMAx (new lines 173, 182, 184, 228)
Comment #9
Section 2.2 does not reference miRNAs that are in circulation and it is unclear how it is proposed that miRNAs present in the feces are entering circulation and impacting the brain.
Authors’ reply
We thank the Reviewer for this comment. We add some lines to clarify that fecal miRNAs can be considered as circulating miRNAs, as those found in blood, saliva, urine, and that most of these miRNAs were detected in fecal EVs derived by both the intestinal microbiota and the host. Therefore, the correlation between altered fecal miRNAs expression and brain dysfunction could depend on the systemic activity of these miRNAs (new lines 261-267).
Comment #10
Line 195, “further studies are necessary to confirm the efficacy of miRNAs as treatments”: Is this a stagnant area, or are studies on the way? Are there studies using EVs themselves as miRNA transport as a treatment?
Authors’ replay
We thank the Reviewer for this consideration. We added some clarifications (new lines 269-273)
Comment #11
Lines 227-228: This statement should be rephrased. miRNAs primarily function within their tissue of origin. Additionally some work indicates they also function in intercellular communication through EVs. This statement is not citing a primary literature article, so it is hard to say the evidence that backs it up. Authors’ reply
We thank the Reviewer for this comment. We rephrased the statement (new line 309).
Comment #12
Line 249 and 26: NSCLC – typo
Author replay
We thank the Reviewer. We corrected the typo (new lines 330, 345)
Comment #13
Line 274: How would identifying newer miRNAs lead to further understanding of the understanding of the lung-brain? Are there any specific pathways of interest? What would these studies look like?
Authors’ replay
We thank the Reviewer for this comment. We added an example for the study of the role of miRNAs in the formation of BM from lung cancers, that is usually the methodology used to investigate how miRNAs altered expression may affect lung/peripheral organ-brain axis pathways (new lines 357-367).
Comment #14
Line 369: word choice should be modified to inter-organ
Author replay
We thank the Reviewer for this suggestion. We modified the word as suggested (new line 466)
Comment #15
Line 440-443: How does exercise relate to these changes in 'myomiRs" in skeletal muscle.
Author replay
We thank the reviewer for this comment. Exercise significantly influences the expression of myomiRs, which play key roles in regulating muscle function, homeostasis, and inter-organ communication, including signaling to the brain. MyomiRs such as miR-1, miR-133, and miR-206 are particularly responsive to physical activity, and their expression is tightly regulated during exercise. We added some lines referred to this concept when we first talked about the influence of exercise on myomiR expression patterns (new lines 476-485).
Comment #16
Define POMC in line 490
Author replay
We thank the reviewer for this suggestion. We added the definition for POMC (new lines 607).

Reviewer 2 Report
Comments and Suggestions for Authors
This manuscript titled “ Brain-periphery axes: the potential role of extracellular vesicles-delivered miRNAs” provides a comprehensive review regarding the role of extracellular vehicles (EVs) and miRNAs in mediating brain-periphery communication. This review provides valuable insights for advancing the treatment and management of neurological and systemic diseases. However, I have a few minor concerns which needs to be addressed to improve the overall quality of the manuscript.
1- Discuss all the limitations like challenges in EV isolation and characterization.
2- Are there ethical or regulatory considerations for translating EV-based therapies to humans?
3- Are there any common miRNAs that could serve as biomarkers for multiple axes discussed in this manuscript?
4- Explain in detail how EV-mediated miRNA transfer differs from other extracellular mechanisms (e.g. neurotransmitters)?
5- What are the structural and molecular properties that allow EVs to cross the Blood-Brain barrier?
Author Response
Reviewer#2
This manuscript titled “Brain-periphery axes: the potential role of extracellular vesicles-delivered miRNAs” provides a comprehensive review regarding the role of extracellular vehicles (EVs) and miRNAs in mediating brain-periphery communication. This review provides valuable insights for advancing the treatment and management of neurological and systemic diseases. However, I have a few minor concerns which need to be addressed to improve the overall quality of the manuscript.
Comment#1
Discuss all the limitations like challenges in EV isolation and characterization.
Authors’ reply
We thank the Reviewer for this suggestion. We added a brief description of the EVs isolation and characterization techniques (new lines 697-704).
Comment#2
Are there ethical or regulatory considerations for translating EV-based therapies to humans?
Authors’ reply
We thank the Reviewer. We cited a work discussing the ethical concerns of using EV-based therapies on humans (new lines 133-136)
Comment#3
Are there any common miRNAs that could serve as biomarkers for multiple axes discussed in this manuscript?
Authors’ reply
We thank the Reviewer. The intent of the review is also to find a systematic approach in searching for miRNAs implicated in diseases involving multiple organs and thus the possibility of finding new biomarkers. Unfortunately, however, with current studies it is not yet possible to identify them certainly but it is possible only make predictions that should be confirmed experimentally.
Comment#4
Explain in detail how EV-mediated miRNA transfer differs from other extracellular mechanisms (e.g. neurotransmitters)?
Authors’ reply
We thank the Reviewer. We briefly discussed the difference between EVs’ cargo release, and that of other molecules such as neurotransmitter and hormones (new lines 86-109)
Comment#5
What are the structural and molecular properties that allow EVs to cross the Blood-Brain barrier?
Authors’ replay
We thank the Reviewer. We briefly discussed this issue (new lines 117-124).

Reviewer 3 Report
Comments and Suggestions for Authors
The authors provide a systemic review on Brain-periphery axes with specific focus on miRNA. This is interesting to readers.
However, the authors have missed some key brain-periphery axes, such as immuno-brain axes and kidney-brain axes.
In immuno-brain axes, some exosome miRNA play roles, such as miR-124, miR-155, miR-21, 29, and 146a. In kidney-brain axes, miR223, 92a, 210. Furthermore, miR-122, 137, 21, and 124 have therapeutic significance.
In the gut-brain axes, the authors have missed the pathological roles of synuclein seed in PD pathogenesis. The authors should add more discussion on it.
Author Response
Reviewer#3, Comment#1
The authors provide a systemic review on Brain-periphery axes with specific focus on miRNA. This is interesting to readers.
Authors’ reply
We thank the Reviewer for this comment.
Reviewer#3, Comment#2
However, the authors have missed some key brain-periphery axes, such as immuno-brain axes and kidney-brain axes.
In immuno-brain axes, some exosome miRNA play roles, such as miR-124, miR-155, miR-21, 29, and 146a. In kidney-brain axes, miR223, 92a, 210. Furthermore, miR-122, 137, 21, and 124 have therapeutic significance.
Authors’ reply
We thank the Reviewer for this suggestion. We added two more paragraphs briefly describing the immune-brain axis and the kidney-brain axis (new lines 619-694), citing some of the suggested miRNAs. In agreement we modified the graphical abstract, Figure 1, the keywords and the conclusion.
Reviewer#3, Comment#3
In the gut-brain axes, the authors have missed the pathological roles of synuclein seed in PD pathogenesis. The authors should add more discussion on it.
Authors’ replay
We thank the Reviewer for this comment. We added some information on the pathological roles of alpha synuclein in PD in gut-brain communication (new lines 158-167).

Round 2
Reviewer 1 Report
Comments and Suggestions for Authors
The authors have thoughtfully revised and added to their review making it a strong manuscript that contributes to readers general understanding of the research area.
Reviewer 3 Report
Comments and Suggestions for Authors
The authors have addressed all concerns of reviewers properly. The quality of the review has been significantly improved after revision. So I suggest accepting of the article in the current form.